# Plasticity of High-Density Neutrophils in Multiple Myeloma is Associated with Increased Autophagy Via STAT3

**DOI:** 10.3390/ijms20143548

**Published:** 2019-07-19

**Authors:** Fabrizio Puglisi, Nunziatina Laura Parrinello, Cesarina Giallongo, Daniela Cambria, Giuseppina Camiolo, Claudia Bellofiore, Concetta Conticello, Vittorio Del Fabro, Valerio Leotta, Uros Markovic, Giuseppe Sapienza, Alessandro Barbato, Silvia Scalese, Daniele Tibullo, Maria Violetta Brundo, Giuseppe Alberto Palumbo, Francesco Di Raimondo, Alessandra Romano

**Affiliations:** 1Department of Surgery and Medical Specialties, University of Catania, 95123 Catania, Italy; 2Division of Hematology, Azienda Ospedaliera Policlinico e Vittorio Emanuele di Catania, 95123 Catania, Italy; 3Department of Medical, Surgical Sciences and Advanced Technologies “G. F. Ingrassia”, University of Catania, 95123 Catania, Italy; 4CNR-IMM, Ottava Strada n.5, 95121 Catania, Italy; 5Department of Biomedical and Biotechnological Sciences, University of Catania, 95123 Catania, Italy; 6Department of Biological, Geological and Environmental Science, University of Catania, 95124 Catania, Italy

**Keywords:** neutrophil, autophagy, IFN-γ, STAT-3, ruxolitinib

## Abstract

In both monoclonal gammopathy of uncertain significance (MGUS) and multiple myeloma (MM) patients, immune functions are variably impaired, and there is a high risk of bacterial infections. Neutrophils are the most abundant circulating leukocytes and constitute the first line of host defense. Since little is known about the contribution of autophagy in the neutrophil function of MGUS and MM patients, we investigated the basal autophagy flux in freshly sorted neutrophils of patients and tested the plastic response of healthy neutrophils to soluble factors of MM. In freshly sorted high-density neutrophils obtained from patients with MGUS and MM or healthy subjects, we found a progressive autophagy trigger associated with soluble factors circulating in both peripheral blood and bone marrow, associated with increased IFNγ and pSTAT3S727. In normal high-density neutrophils, the formation of acidic vesicular organelles, a morphological characteristic of autophagy, could be induced after exposure for three hours with myeloma conditioned media or MM sera, an effect associated with increased phosphorylation of STAT3-pS727 and reverted by treatment with pan-JAK2 inhibitor ruxolitinib. Taken together, our data suggest that soluble factors in MM can trigger contemporary JAK2 signaling and autophagy in neutrophils, targetable with ruxolitinib.

## 1. Introduction

In multiple myeloma (MM), the accumulation in the bone marrow (BM) of neoplastic plasma cells (PCs) secreting a monoclonal immunoglobulin (Ig) leads to anemia, bone pain, renal impairment, hypercalcemia, and infections [1]. Virtually in all MM patients, the symptomatic phase is anticipated by a spectrum of indolent clinical variants known as monoclonal gammopathy of uncertain significance (MGUS) and smoldering MM [2].

In both MGUS and MM patients, immune functions are variably impaired, and there is a high risk of bacterial infections [3].

Neutrophils are the most abundant circulating leukocytes and constitute the first line of host defense. Under physiological conditions, when inflammation is triggered due to, for example, exposure to bacteria and neo-antigens, neutrophils egress from the BM to the circulation and migrate to tissues to execute their immunological functions, thus representing one of the main populations of cells involved in inflammation [4,5]. Thanks to a plethora of integrins and molecules expressed on their surface, neutrophils can interact with cells belonging to the immune system (e.g., T and B cells, monocytes, macrophages, dendritic cells, and natural killer cells) or involved in its regulation like endothelial and mesenchymal stem cells in the hematopoietic niche [6,7,8].

There is an increasing interest for neutrophils in solid cancer where two types of neutrophils (tumor-associated neutrophils, TANs) have been characterized by differential morphology, phenotype, and function, including angiogenesis promotion [9], based on observations of tumor-bearing mice and patients [10]. Clinically relevant in several settings, including solid and hematological cancers, the ratio between the absolute number of neutrophils (ANC) and lymphocytes (ALC) can predict outcome and overall survival, as surrogates of both chronic inflammation and immune-suppressive status [11,12,13].

Recent evidence has suggested that in several pathological contexts, including cancer, neutrophils can shift their cell skills in response to environmental changes, overcoming the traditional vision of terminally differentiated cells [14,15]. The picture of the emerging functional and phenotypic heterogeneity of neutrophils is further enriched by the notion that in murine models, myeloid-derived suppressor cells with overlapping neutrophil phenotypes are increased and associated with outcome, as shown also in MM [16,17]. Moreover, in the cancer-induced myelopoiesis, neutrophil maturation can be affected by cancer-related cytokines triggering RORC1 signaling [18]. Our group previously showed that MM-BM mesenchymal stromal cells educated by MM-released soluble factors can induce the expansion of Neutrophils and polymorphonucler myeloid-derived suppressor cells (PMN-MDSC) with increased angiogenic and immune-suppressive activities [8].

Autophagy is an important mechanism that regulates the late stages of differentiation of myeloid precursors toward granulocytes and monocytes [19,20], assuring bioenergetics and proliferation rate of neutrophils in the bone marrow, blood, and spleen, without affecting their functionality, including apoptosis and migration [21,22]. Autophagy is required to fuel Mitochondrial Oxidative Phosphorylation System (OXPHOS) with fatty acids through lipophagy and lipid droplet generation, resulting in a swift shift from glycolysis to OXPHOS, especially in the late stages of neutrophil differentiation [21].

In MM, neoplastic PCs are addicted to autophagy, as required to compensate the high protein biosynthesis rate, stress of endoplasmic reticulum, and mitochondria dysfunction [23,24]. However, autophagy inhibitors such as chloroquine have been tested in preclinical models and phase I–II trials with dismal results [25]. We hypothesized that autophagy can affect the hematopoietic niche within the bone marrow in different ways and represents a compensatory pathway in the surrounding non-neoplastic cells.

Since little is known about the contribution of autophagy in the neutrophil function of MGUS and MM patients, we investigated the basal autophagy flux in freshly sorted neutrophils of patients and tested the plastic response of healthy neutrophils to soluble factors of MM.

## 2. Results

### 2.1. Changes in the Neutrophil Phenotype are Associated with Autophagy Triggering in Monoclonal Gammopathy Of Uncertain Significance MGUS and Multiple Myeloma (MM) Patients

High-density neutrophils isolated from MM patients had hyposegmented nuclei, with reduced lobularity than those isolated from MGUS and healthy subjects (Figure 1A–L). Changes in morphology were associated with differences in function, since MM high-density neutrophils (HDNs) had more autophagosomes and autolysosomes than healthy HDNs as detected by electron microscopy (Figure 1M).

Different from PMN-MDSCs isolated in the upper ring after Ficoll separation, MM- and healthy-derived HDNs did not show any significant difference in LOX-1 expression, a marker of neutrophil immaturity related to immune-suppressive activity. As previously reported in other cancer settings [17], LOX-1 was increased in MM-derived CD15^+^ low-density neutrophils (LDN)s but not in CD15^+^HDNs (Figure 2A–D).

In the time-course shown in Figure 2, after 96 h of culture in vitro, MM-HDNs were still alive, as opposed to healthy-derived HDNs, as detected by two independent series of experiments using 7-AAD/AnnexinV (Figure 2E–G) assay by flow-cytometry and lactate dehydrogenase (LDH) cell viability assay (Figure 2H). The production of reactive oxygen species, as detected by flow cytometry measuring the mean fluorescence intensity of DCFDA staining, was significantly higher in MM than MGUS/healthy HDN (Figure 2I). Both healthy and MM-HDNs produced extracellular traps (NETs) after 48 h of cell culture, in a process that counter-balanced neutrophil apoptosis [26] (Figure 2L). 

According to the electronic microscopy images, we saw an increased number of acidic vesicular organelles (AVOs) in MM-HDN compared to healthy-derived HDNs, as detected by flow-cytometry (respectively, 16.3 ± 1.8 versus 6.3 ± 1.3 versus 2.3 ± 0.8%, *p* = 0.001, Figure 3A), associated with increased autophagy flux as detected by immunoblotting of endogenous unconjugated LC3I to lipid conjugated LC3-II after treatment with bafilomycin for 3 h (Figure 3B). Conversely, immunoblots of p62/SQTM1 autophagy receptors disclosed a reduced amount of detectable protein in MM-HDNs compared to healthy-HDNs (Figure 3C).

Autophagy-related genes were higher in MM-HDN and MGUS than healthy HDN, more than twofold in LC3B (*p* < 0.001) and p62/SQSTM1 (*p* = 0.001) expression, while ATG10 (*p* = 0.02) and GABARAP (*p* = 0.001) expression was increased only in MM; MGUS-HDN and other autophagy receptors, like OPTN, were not affected (Figure 3D).

### 2.2. MM-Related Soluble Factors Can Trigger Autophagy in Neutrophils

Then, we tested if MM-related soluble factors could trigger autophagy in healthy neutrophils. To this end, HDNs isolated from three individual healthy donors were incubated for 3 h with sera obtained from peripheral blood (PB) from 3 healthy donors, and they were compared to MM patient sera obtained from bone marrow (BM) and PB; autophagy triggering was measured with AVO by flow-cytometry (Figure 4A). Soluble factors in MM sera could increase the AVO percentage in healthy HDN (PB sera: 139% ± 11%, BM sera 140% ± 2%), even if there was not any significant difference between matched BM and PB obtained from the same patient (Figure 4A).

Healthy HDNs cultured for 3 h with MM but not MGUS PB sera had more AVOs than those cultured in the presence of allogeneic healthy donor PB serum (respectively, 169 ± 11 versus 98 ± 4, *p* = 0.04; given 100% incubation with healthy donors). The effect was reverted by treatment with 200 nM of the autophagy inhibitor bafilomycin (169 ± 11 versus 111 ± 4, *p* = 0.009, Figure 4B). Data were confirmed by immunofluorescence detection of LC3B puncta in the presence or absence of bafilomycin (Figure 4C).

### 2.3. Autophagy Induction in Neutrophils is Associated with IFNγ Signaling and Downstream STAT3 Activation

In order to establish the contribution of the microenvironment—characterized by increased levels of inflammatory cytokines such as IL-6, a key-player of growth and survival factor for myeloma cells [28,29]—and to induce changes in the HDN phenotype, we evaluated the engagement of the JAK2-STAT3 pathway in autophagy triggering in neutrophils.

In MM-HDN, phosphorylation of STAT3-S727 was higher than MGUS and healthy HDN (respectively 18.7 ± 3.8 versus 5.3 ± 0.3 versus 3.4 ± 0.5 in MM, *p* = 0.02, Figure 5A), and was associated with increased protein synthesis of IFNγ, which was higher in MM than MGUS (7.5 ± 0.2 versus 3.8 ± 0.4, *p* = 0.02) and healthy HDN (7.5 ± 0.2 versus 2.7 ± 0.06, *p* < 0.0001, Figure 5B).

Conversely, in HDN isolated from three individual healthy donors, incubated for 3 h with completed RPMI or conditioned MM media (obtained from human MM cell lines MM1.s or U266), the percentage of AVO increased (*p* = 0.001, Figure 5C) and was associated with an increase of STAT3 phosphorylation on S727 (*p* = 0.01, Figure 5D). In the presence of 500 nM of the pan-JAK inhibitor ruxolitinib, reduction in STAT3 phosphorylation on S727 (128% ± 4% versus 103% ± 3%, *p* = 0.001, Figure 5C) was associated with reduced autophagy (AVO percentage 169 ± 12 versus 74 ± 1, *p* = 0.001, Figure 5D), confirming the association between availability of pSTAT3 and autophagy.

## 3. Discussion

Our findings disclosed an increase in autophagy flux and JAK-2/STAT3 activation status in MM high-density neutrophils, which could contribute to their promotion of pro-inflammatory and survival signals within the plasma cell niche. Our work has focused on high-density neutrophils, given the functional overlap between low-density neutrophils and granulocytic myeloid suppressor cells in cancer, which is not completely explained in the myeloma setting [10,16,17,18,30].

Autophagy is a major innate immune defense to control intracellular pathogens (e.g., mycobacterium tuberculosis, viral virulence gene products) elicited by cytokines, like IFN-γ, or pattern recognition receptors (TLRs and nucleotide-binding oligomerization domain-like receptors) [31]. In neutrophils, autophagy is required to mediate antimicrobial capacity in response to various stimuli, derived from the inflammatory microenvironment, to deliver disease-related proteins to the extracellular space within the NET scaffold [20].

Second, we found that in the evolution from MGUS through MM, HDN progressively increased the production of IFN-γ. IFN-γ is a crucial T-helper 1 cytokine, which acts as an important bridge between innate and adaptive immunity and is involved in many acute and chronic pathologic states. In cancer, IFN-γ reduces cell growth and favors non-apoptotic cell death in hematological [32] and solid settings [33], via autophagy induction [34].

Very little is known about how IFNγ is induced in neutrophils. In non-neoplastic conditions, like *Streptococcus pneumoniae* infection in mice, neutrophil IFN-γ production is under control of toll-like receptor engagement via MYD88, but not the CD11b/CD18 complex, and induces transcription of target genes (members of nonreceptor Src kinase family, Hck, Fgr, and Lyn), which are critical for the acute inflammatory response and initiation of immune response [35]. The Src family kinases Hck, Fgr, and Lyn are recruited by IL-6, typically increased in the MM microenvironment, and for this reason they have been proposed as a novel target to augment the activity of current MM therapies [36,37].

Cells of a myeloid lineage utilize Hck, Fgr, and Lyn to mediate Fc gamma receptor (FcγR) signaling and regulate events associated with the engagement and activation of a variety of receptor classes, including chemokine receptors, adhesion molecules, and lectins [38,39]. For example, Hck, Fgr, and Lyn are required to generate the inflammatory environment in vivo and to release pro-inflammatory mediators from neutrophils and macrophages in vitro, but not for their intrinsic migratory ability [38]. MM neutrophils display increased levels of Hck, Fgr, and Lyn (Parrinello, unpublished data), but their involvement in mediating tumor-associated neutrophil dysfunction has never been investigated.

In eukaryotes, JAK2/STAT3 engagement is associated with significant intracellular changes, through which the expression of multiple cytokines and growth factors, cell proliferation, differentiation, and apoptosis occur, which are relevant for cancer progression and drug resistance [40,41]. In CD138^+^ neoplastic plasma cells, despite the absence of mutations, overexpression of JAK2 and STAT3, due to miRNA-375 promoter hypermethylation, was reported in almost half of MGUS and MM patients [42].

In the presence of pro-inflammatory cytokines, such as IL-6 and TNF-α, JAK2 can bind with the relevant receptors, thus activating the tyrosine residue of all downstream target proteins (Figure 6). To exert biological effects, phosphorylation of JAK2/STAT3 should be rapidly completed; indeed, it disappears within 6 h, mediating short-term events [43]. As a substrate of JAK kinase, STAT, once being activated, can pass through the nuclear membrane in the form of polymers, such as dimers or tetramers, to specifically bind to response elements on DNA to initiate the transcription of downstream target genes, including those coding for autophagy machinery.

Soluble factors such as IFN-γ, IL-4, IL-13, and IL-6 are associated with autophagic regulation, even if in opposite directions, based on cell type and experimental settings. In macrophages, IFN-γ accelerates both the formation and the maturation of autophagosomes via the JAK1/2 signaling pathway [44], leading to STAT-1 independent transcripts, like caspase 7-like Lice-2, IL-1β, Daax, and the serine protease PIM-1, which coordinate apoptotic/pyroptotic responses possibly in relation to autophagic stimuli [45]. Other groups reported that the IL-6/JAK2 signaling pathway inhibits autophagy, via p-STAT3, which activates Bcl-2 to regulate the expression of Beclin 1 and VPS34 [46], while in pancreatic cancer, IL-6 induces mitochondrial localization of p-STAT3 at Ser727, upregulating autophagy flux [47].

Both cytoplasmic STAT3 and nuclear p-STAT3 can affect autophagy. In cancer cells, nuclear p-STAT3-S705 can regulate the transcription of several autophagy-related genes such as BCL2 family members (e.g., BECN1, PIK3C3, CTSB, CTSL, PIK3R1, HIF1A, and BNIP3) and microRNAs with targets of autophagy modulators [47]. However, STAT3 can trigger autophagy also without phosphorylation usually detected in cancer, in line with our observations. Because of the interaction between the SH2 domain of cytoplasmic, unphosphorylated STAT3 and the catalytic domain of the eIF2α kinase 2 (GCN2), eIF2α can be hyperphosphorylated and induce an adaptive response that converges on autophagy induction [48]. Cytoplasmic p-STAT3-ser727 constitutively inhibits mitophagy by sequestering GCN2 and its downstream effectors, preserving mitochondria from being degraded when macro-autophagy is triggered by starvation [49], a pathway that could be relevant to preserve neutrophils from apoptosis, thus explaining the long-term survival in culture of MM-HDN.

Altogether, our data established that IFN-γ, produced by MM-HDN in response to soluble factors released by neoplastic plasma cells, can trigger the JAK2/STAT3 pathway to increase autophagy, contributing to long-term survival. The pan-JAK2 inhibitor ruxolitinib, by reducing levels of phosphorylation of STAT-Ser-727 in neutrophils that have only a few mitochondria, can globally reduce autophagy induction and potentially revert the pro-inflammatory contribution of neutrophils to MM survival. An encouraging report to this end has been recently published [50], disclosing that the selective JAK1 inhibitor INCB052793 can inhibit cell viability of MM cells when used alone or in combination with proteasome inhibitors and glucocorticosteroids. Further studies are required to address the clinical benefit of JAK/STAT inhibitors in reducing the intracellular effects of pro-inflammatory cytokines released by MM cells in the microenvironment.

## 4. Materials and Methods

### 4.1. Patients and Controls

Between January and April 2018, high-density neutrophils (HDNs) were derived from 10 newly diagnosed MM and 10 MGUS patients. Patients were free from immune-mediated diseases and acute or chronic viral infections to avoid any interference on immune-regulatory mechanisms. All MGUS patients had a stable chronic disease with at least 2 years of follow up. Ten healthy subjects (age >45 years) were recruited in the study as controls.

None of the recruited patients was receiving medical treatments that could have an impact on their immune condition. No subjects refused authorization to use their medical records for research, and all provided their consent according to the Declaration of Helsinki.

### 4.2. Isolation of High-Density Neutrophils (HDNs)

Whole blood (40 mL) was collected from healthy volunteers, MGUS, and MM patients in vacutainer tubes containing the anticoagulant, potassium EDTA, and diluted 1:1 with dextran 3% for two hours to obtain plasma enriched with white cells. Peripheral blood mononuclear cells (PBMCs) were then isolated by the standard method of density gradient centrifugation using Ficoll-Paque (Pharmacia LKB Biotechnology, NJ, USA). The resulting interphase layer from the gradient was diluted and washed twice with Dulbecco’s phosphate-buffered saline (PBS) (Celbio) to obtain PBMC from the top and neutrophils from the bottom.

The pellet obtained after centrifugation of PB on Ficoll, containing erythrocytes and high-density neutrophils (HDNs), was subjected to hypotonic lysis (155 mM NH_4_Cl, 10 mM KHCO_3_, 0.1 mM EDTA, pH 7.4) for 15 min on ice. After washing, cells were further immune-magnetically sorted using the EasySep human neutrophil Isolation kit (StemCell Technology, cat #17957). HDN purity and viability were checked by morphology and flow cytometry (Figure 1). HDNs with purity and viability of more than 95% were used for further assays.

### 4.3. Evaluation of Morphology in HDN

Sections from the same samples were used for cytochemistry and electron microscopy. Morphological staining was carried out on cytospins (5 μm thick) using hematoxylin/eosin (H/E, Bio-Optica). Nonspecific binding sites for immunoglobulins were blocked by incubation for 1 h with normal goat serum (NGS) in PBS (1:5).

For conventional electron microscopy (EM), cells were pelleted and processed as described. Briefly, ultrathin sections were contrasted with uranyl acetate and lead citrate and observed with a JEOL JEM 2010 transmission electron microscope (Houghton, MI, USA) with a LaB6 thermoionic source operating at an acceleration voltage of 200 kV. In conventional EM analyses, micrographs of randomly selected cells were digitalized.

### 4.4. Immune-Phenotype of HDN

One hundred thousand HDNs were separated using a combination of physical and immune-magnetically based methods, as described above, and they were stained with the following antibodies from Beckman Coulter: CD15-FITC (clone80H5), CD11b-PE (clone bear-1), CD14-PC7 (clone RMO52), and CD45-APC (clone J33); in selected experiments, to exclude contamination of MDSC, HLA-DR-ECD (clone Immu357), CD45-PC5 (clone J33), and LOX-1-APC (Biolegend, clone 15C4) were tested as well.

Purity and immune-phenotype of HDNs were assessed by forward and 90° light scatter parameters after staining with the following antibodies from Beckman Coulter: CD14-PC7 (clone RMO52), CD15 FITC (clone 80H5), and HLA-DR ECD (clone Immu-357). The purity of HDN, identified as CD15+CD14-HLADR cells, was over 95% (Figure 1A). Functional activity was detected looking at the intracellular levels of IFN-γ and phosphorylated STAT-3 using CD15-FITC (clone 8H05) and IFN-γ-PE (clone 45.15) monoclonal antibodies from Beckman Coulter, STAT3pS727Vio515 (clone REA324) from Mylteni, and respective isotypic controls, following the appropriate instructions of the manufacturer, after stimulation with the Duractive Stimulation kit (Beckman Coulter), which contained phorbol-myristate-acetate (PMA) and ionomycin as cellular activators as well as brefeldin A to block the secretion from Golgi apparatus, for three hours at 37 °C. After that, cells were fixed and permeabilized using the PerFix-nc-kit (Beckman Coulter, Indianapolis, United States), and 50,000 events were acquired using a Navios flow cytometer. Results were expressed as mean fluorescence intensity (MFI) corrected for values of nonspecific binding. For more robust statistical evaluation, MFI values were converted to a resolution metric such as the RD, defined as (Mediantreatment-Mediancontrol)/(rSDtreatment+rSDcontrol), to further perform *t*-tests to compare results of different experiments and runs.

### 4.5. Ex Vivo Treatment of HDN with MM-Conditioned Media or Sera Derived from MGUS-MM Patients

For sera collection, blood was centrifuged for 10 minutes at 1600× *g* and then for 10 min at room temperature, and the supernatant was saved at −80 °C for a maximum of 2 months. Two cells lines, MM1.s and U266 human myeloma cell lines (HMCLs), were kindly provided by Prof. Tassone (Magna Grecia, University of Catanzaro), which were previously validated using sequencing and phenotypic characterization. Cells were plated 72 h prior to collection of conditioned media, which was then filtered using a 0.2 μm syringe filter and diluted to the appropriate concentration with complete RPMI.

Volunteer-derived HDNs were incubated with 5% CO_2_ at 37 °C at a concentration of 100 × 10 ^4^/mL with sera obtained from 8 MGUS and 8 MM patients (matched for sex and age) or conditioned media to evaluate changes in autophagy flux after 3, 6, and 24 h. In selected experiments, treatment with 50–200 nM bafilomycin purchased by Sigma (St. Louis, MO, USA) was performed to inhibit autophagy. At the end of treatments, cells were collected, and AVOs, p-STAT-3-S727, and IFN-γ were detected by flow cytometry as previously described.

### 4.6. Immunofluorescence

Patient-derived neutrophils for immunofluorescence analysis of LC3B conversion were incubated overnight in a humid chamber at 4 °C with primary antibody anti-LC3 (anti-rabbit, Sigma-HPA003595). The next day, cells were washed three times in PBS for 5 min and incubated with secondary antibodies at a dilution of 1:200 and PE-conjugated (anti-rabbit, Sigma-F0382) for 1 h at room temperature. The slides were mounted with medium containing DAPI (4′,6-diamidino-2-phenylindole, Santa Cruz Biotechnology, Santa Cruz, CA, USA) to visualize nuclei. Coverslips were observed using a Zeiss Axio Imager Z1 Microscope with Apotome 2 system (Zeiss, Milan, Italy), equipped with an AxioCam camera (Zeiss, Jena, Germany). After first observation of the tissue sections under a 20× objective, we identified five fields with the largest number of immunostained cells. Then, using 40× oil-immersion objective, the immune-positive cells were counted in each one of these fields. The numerical aperture was 1.35 (40× lens), and images of HDNs were deconvoluted with SoftWorx 3.5.0 (Applied Precision, Bratislava, Slovakia). Fluorescence intensity was quantified in an automated fashion with IN Cell Investigator software (GE Healthcare, Piscataway, NJ, USA).

### 4.7. Expression of Autophagy-Related Genes by qRT-PCR

Total RNA was extracted from neutrophils using Trizol reagent and quantified using a UV spectrophotometer. One microgram of total RNA (in 20 μL reaction volume) was reverse-transcribed in cDNA using reverse-transcriptase (Roche Diagnostic Corp., Indianapolis, IN, USA) and oligo-dT primers in a standard reaction. The quantitative real-time polymerase chain reaction (RT-PCR) of the resultant cDNA was performed using a LightCycler (Roche),with 300 nM primers designed specifically for the transcripts of hATG10 FW: TACGCAACAGGAACATCCAA; hATG10 RV: AACAACTGGCCCTACAATGC; hATG12 FW: CAGTCGCTACTTCCGCTCTCGAG; hATG12 RV: AAACAATGTTCTGAGGCCACAAG; hMAP1LC3B FW: GAACGATACAAGGGTGAGAAGC; hMAP1LC3B RV: AGAAGGCCTGATTAGCATTGAG; GABARAP FW: ACACTGACAATTTCATCCCG; hGABARAP RV: GCCTTTCCCA TCCTGCTGTA; hSQSTM1 FW: GGGGCGCCCTGAGGAACAGA; hSQSTM1 RV: CCTGGTGAGCCAGCCGCCTT; hOPTN FW: TTCGGCCTGGACAGAGAAAC; hOPTN RV: TGCCTTCTCTGCTTGTAGCC, according to the gene manufacturer’s recommended protocol (Applied Biosystem). Each reaction was run in triplicate. Samples were quantified accordingly (LightCycler analysis software, version 3.5) using the housekeeping gene GAPDH (Fw: TCCTGTTCGACAGTCAGCCGCA, Rv: GCGCCCAATACGACCAAATCCGT) as standard.

### 4.8. Acridine Orange Staining and Flow Cytometric Analysis

Formation of acidic vesicular organelles (AVOs) was quantified by acridine orange staining to monitor autophagy triggering. Cells were washed with phosphate-buffered saline (PBS), then acridine orange (0.5 mg/mL, Invitrogen, Waltham, MA, USA) was added for 15 min prior to collection, and cells were analyzed by flow cytometry, as previously described.

### 4.9. Reactive Oxygen Species (ROS) Detection

To detect the production of reactive oxygen species (ROS), healthy and MM-HDNs were treated with 5 μM of 2′,7′-Dichlorodihydrofluorescein diacetate (DCFH-DA, Sigma-Aldrich, prepared in ethanol and kept at −20 °C) at 37 °C and 5% CO2 for 30 min in the dark according to the instructions of the manufacturer. Samples were analyzed by cytometry using a 488 nm laser for excitation and detection at 535 nm.

### 4.10. Lactate Dehydrogenase (LDH) Activity Assay for Cell Viability Detection

The healthy and MM-HDN cell viabilities were evaluated using a lactate dehydrogenase (LDH) activity assay (CytoSelectTM LDH Cytotoxicity Assay Kit, Cell Biolabs, San Diego, CA, USA) following the manufacturer’s instructions. For the assay, 10% Triton X-100 in PBS-treated cells was the positive control (100% toxicity) and untreated live cells were the negative control (0% toxicity). At the end, the percentage of relative cytotoxicity for experimental samples was calculated as: [(ODsample − ODnegative control)/(ODpositive control − ODnegative control)] × 100. Assays were performed on the media of healthy and MM-HDNs after 96 h of cell culture.

### 4.11. Statistical Methods 

Statistical analyses for subject data and functional analyses were performed using Prism GraphPad Software (La Jolla, CA, USA). Two-tailed Student’s *t*-tests and Fischer’s exact tests were used to compare continuous and categorical clinical variables, respectively. Mann–Whitney U test was utilized for non-parametric data; means and standard deviations, median, and IQR for non-symmetric variables were calculated and compared among healthy, MGUS, and MM subjects by use of a two-tailed Student’s t test with Bonferroni correction. An ANOVA test was used to compare means of more than two groups. Descriptive statistics were generated for analysis of results, and *p*-values under 0.05 were considered significant. Qualitative results were summarized in MFI (mean fluorescence intensity) and in percentages.

## References

## Figures and Tables

**Figure 1 ijms-20-03548-f001:**
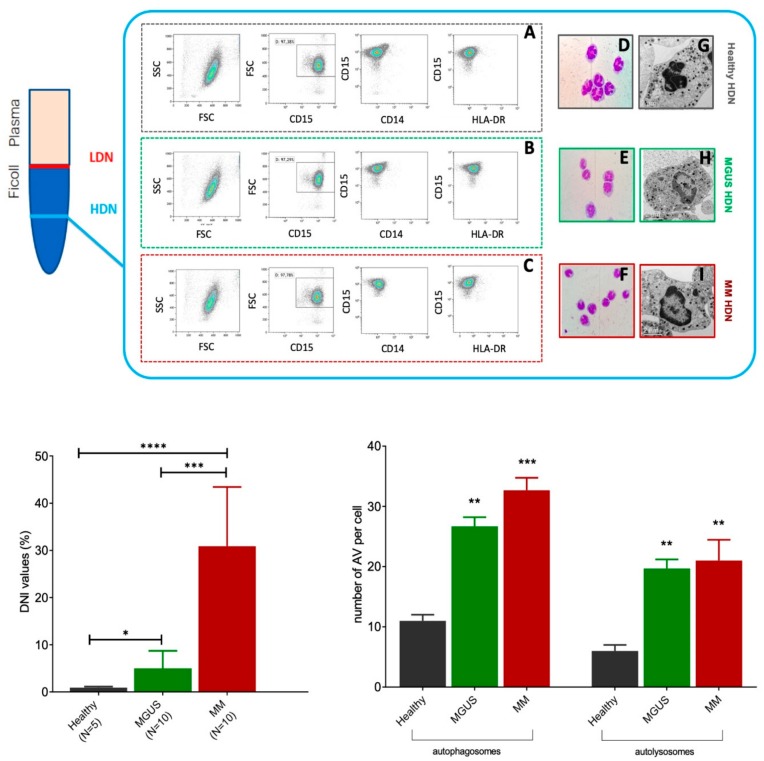
High-density neutrophils (HDNs) isolated from monoclonal gammopathy of uncertain significance (MGUS) and multiple myeloma (MM) patients have distinct morphologies. Mature neutrophils from healthy donors, after peripheral blood centrifugation on Ficoll gradients, typically were in the sediment on top of red cells (high-density neutrophils, HDNs, blue line on the bottom). By contrast, in cancer, immature neutrophils were in the sediment within the mononuclear cell fraction (low-density neutrophils LDNs, red line on the top). HDNs were freshly isolated from peripheral blood of healthy MGUS and MM subjects, and their purity was checked by flow cytometry. In all experiments, HDNs were identified as CD15^+^CD14^−^HLA-DR^−^ cells, with a purity of at least 90%; a representative plot is shown for healthy (**A**), MGUS (**B**), and MM (**C**) subjects. HDN morphology was evaluated by staining with H&E (**D**–**F**, 100× magnification), and features were investigated by optical microscopy (**D**–**F**). Healthy HDN cells have 3–5 segments, while MM and MGUS had lower nuclei segmentation, typically seen in immature cells. **G**–**I** Representative transmission electron microscope images from HDNs isolated from healthy (**G**), MGUS (**H**), or MM (**I**) subjects. (**L**) Delta neutrophils index (DNI) was the DN in leukocyte differentials and was calculated using the following formula: DN = (the leukocyte subfraction assayed in the MPO channel by cytochemical reaction) – (the leukocyte subfraction counted in the nuclear lobularity channel by the reflected light beam), using an electronic cell counter (Siemens) in 5 healthy subjects, 10 MGUS, and 10 MM patients. In the same samples, a 200-cell manual differential count was done on blood smears by a hematologist who was blinded to the automated DN count, as previously reported by other groups [27]. (**M**) By electron microscopy, the number of autophagosomes (identified as clear double-membraned structures) and autolysosomes (identified as single-membraned structures containing degraded cytoplasmic components) were estimated by two independent operators (>20 HDNs per subjects; 5 healthy, 5 MGUS, and 5 MM subjects per group). **** *p* < 0.00001, *** *p* < 0.0001; ** *p* < 0.001; * *p* < 0.05 (Student’s *t*-test).

**Figure 2 ijms-20-03548-f002:**
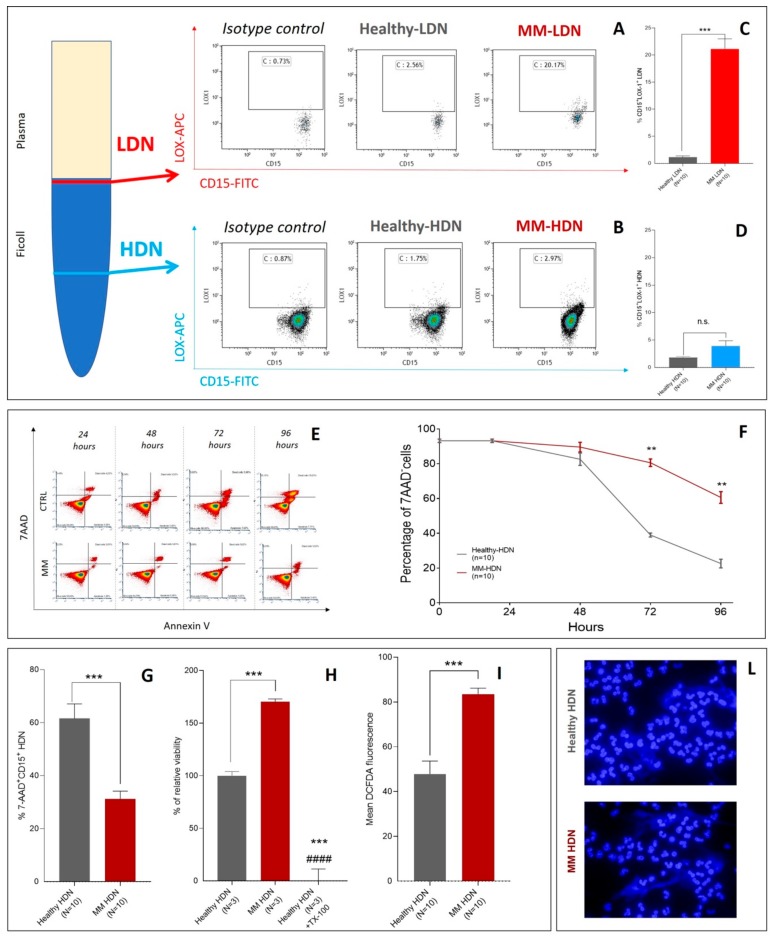
High-density neutrophils of MM patients have distinct phenotypes. Cells were isolated using a density gradient as described in the methods section, and the proportion of LOX-1+ cells was calculated among CD15+ cells by flow cytometry. (**A**) Example of staining with CD15 and LOX-1 antibodies to detect low-density neutrophils (LDN) and (**B**) high-density neutrophils (HDNs) in healthy and MM patients, according to Condamine et al. [17]. (**C**) Proportion of LOX-1 positive LDN and (**D**) LOX-1 positive HDN in peripheral blood of 10 healthy and 10 newly diagnosed MM patients. Individual results for each patient are shown as well as mean and standard deviations. *p* values (*t*-test) are shown. **E**–**G** HDNs were isolated from healthy and MM patients using a density gradient as described in Methods and maintained in culture with 10% FBS complete RPMI media for 96 h. (**E**) Example of time-course (24, 48, 72, and 96 h) of HDN viability as detected by 7-AAD/Annexin V assay in a healthy (top row) and a MM subject (bottom row). (**F**) Changes in the time-course of cell viability for HDN obtained from healthy (grey line) and MM subjects (red line), cultured up to 96 h in 5% CO_2_, 37 °C in complete RPMI medium and +10% FBS. Cell viability was assayed by 7-AAD/Annexin V at each time point indicated. Represented values are mean and standard deviations of five independent experiments using HDNs obtained from 10 healthy and 10 MM subjects. (**G**) Percentage of 7AAD^+^ cells described in F was evaluated at 96 h, showing longer survival in vitro for MM-HDN but not for healthy ones. (**H**) Lactate dehydrogenase (LDH) viability assay on HDNs cultures exposed to increasing concentrations of purmorphamine and cyclopamine (0.01–10 μM). Data are shown as dot plots, and viability is calculated as the percentage of controls assumed as 0%. (**I**) Flow cytometry analysis of intracellular reactive oxygen species (ROS) generation, detected by flow cytometry using Dichlorodihydrofluorescein diacetate (DCFH-DA). (**L**) DAPI staining of DNA revealed the presence of neutrophil extracellular traps (NETs).

**Figure 3 ijms-20-03548-f003:**
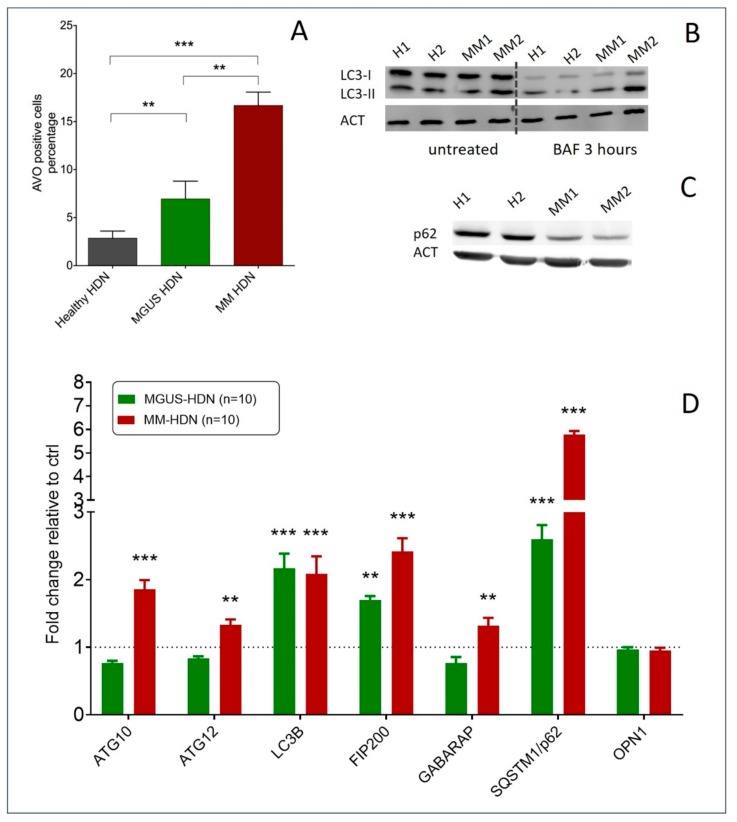
Autophagy is increased in MGUS and MM high-density neutrophils (HDNs). (**A**) The histograms show the average numbers and standard deviations of acidic vesicular organelles (AVOs) quantified by flow cytometry using acridine orange staining, in healthy (*N* = 10), MGUS (*N* = 10), and MM HDNs (*N* = 10) processed within two hours from collection. ** *p* < 0.01, *** *p* < 0.001 (Student’s *t*-test and ANOVA test for multiple comparisons). (**B**) Representative immunoblot of endogenous unconjugated LC3-I to lipid-conjugated LC3-II in healthy and MM HDNs. Cells were treated for 3 h with 50 nM bafilomycin (BAF) or left untreated, lysed in 1% SDS, and analyzed by western blot with anti-LC3 Ab. ACT/actin served as a loading control throughout. (**C**) Representative immunoblot of p62/SQSTM1 autophagy receptor in healthy and MM HDNs. (**D**) Quantitative RT-PCR analysis of fold-changes of transcripts encoding autophagy machinery and responsive proteins in healthy (given 1, dashed line), MGUS, and MM-HDN. *** *p* < 0.0001; ** *p*<0.001 (Student’s *t*-test).

**Figure 4 ijms-20-03548-f004:**
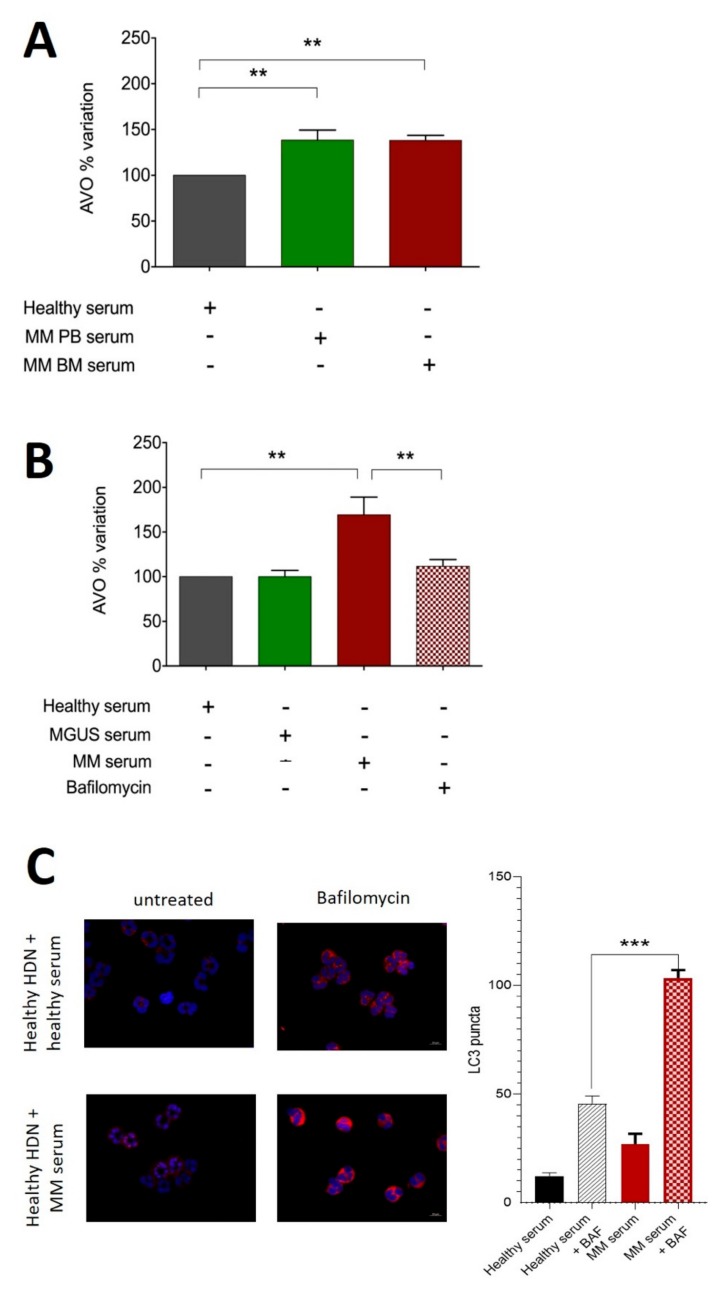
Soluble factors in peripheral blood and bone marrow of MM patients can trigger autophagy in high-density neutrophils. (**A**) HDNs were isolated from healthy donors as described in the Methods section and cultured with sera obtained from MM peripheral blood (PB) and bone marrow (BM) or allogeneic healthy donor for 3 h before being stained with acridine orange to detect the formation of acidic vesicular organelles (AVOs) quantified by flow cytometry. Bars represent mean and standard deviation of 5 independent experiments based on 3 different individual donors and 3 individual MM patients. The experiment in A was repeated using PB sera obtained from MGUS (green bar) or MM (red bar) patients in the presence of the autophagy inhibitor bafilomycin. (**C**) Healthy donor-derived HDNs described in **A**–**B** were seeded on poly-L-lysine-coated slides, fixed with 3.7% formaldehyde for 10 min, and permeabilized with PBS 0.1% Triton X100 10 min at RT. Cells were stained with monoclonal Ab against LC3B (Fk2, Enzo Biochem Inc., Farmingdale, New York, NY, USA; 1:200 1h RT, in red) and DAPI. Coverslips were observed using a Zeiss Axio Imager Z1 Microscope with Apotome 2 system (Zeiss, Milan, Italy) equipped with an AxioCam camera (Zeiss, Jena, Germany). The numerical aperture was 1.35 (40× lens), and images of HDNs were deconvoluted with SoftWorx 3.5.0 (Applied Precision, Bratislava, Slovakia). Fluorescence intensity was quantified in an automated fashion with IN Cell Investigator software (GE Healthcare, Piscataway, NJ, USA). *** *p* < 0.0001; ** *p* < 0.001; (Student’s *t*-test).

**Figure 5 ijms-20-03548-f005:**
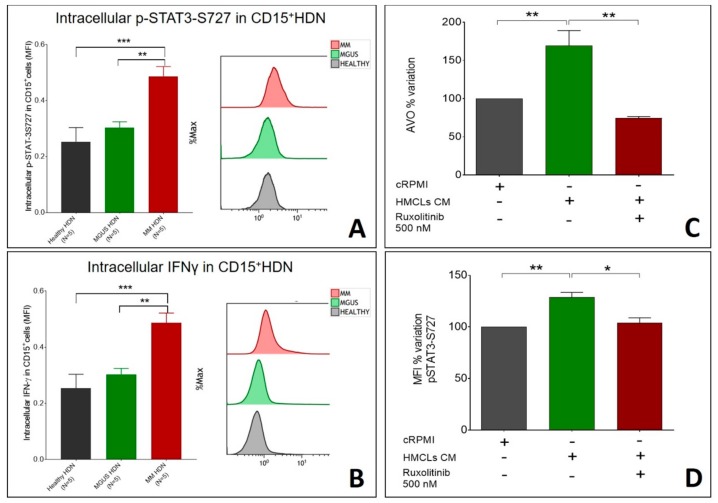
Ruxolitinib can hamper the increase of intracellular IFNγ and p-STAT3-S727 induced by MM soluble factors in HDNs. (**A**) Intracellular interferon-gamma levels, as detected by flow cytometry, in healthy, MGUS, and MM HDN cells are shown, with a representative flow cytometry image for each disease status. (**B**) Intracellular p-STAT3-S7272 levels, as detected by flow cytometry, in healthy, MGUS, and MM HDN cells are shown, with a representative flow cytometry image for each disease status. HDN isolated from healthy subjects were cultured in vitro or 3 h in the presence of conditioned media obtained from MM1.s or U266 cell lines with or without pan-JAK inhibitor 500 nM Ruxolitinib to assess autophagy, detected as the percentage of AVOs after treating with (**C**) and changes in phosphorylation of STAT3 on S727 (**D**). *** *p* < 0.0001; ** *p* < 0.001; * *p* < 0.05 (Student’s *t*-test).

**Figure 6 ijms-20-03548-f006:**
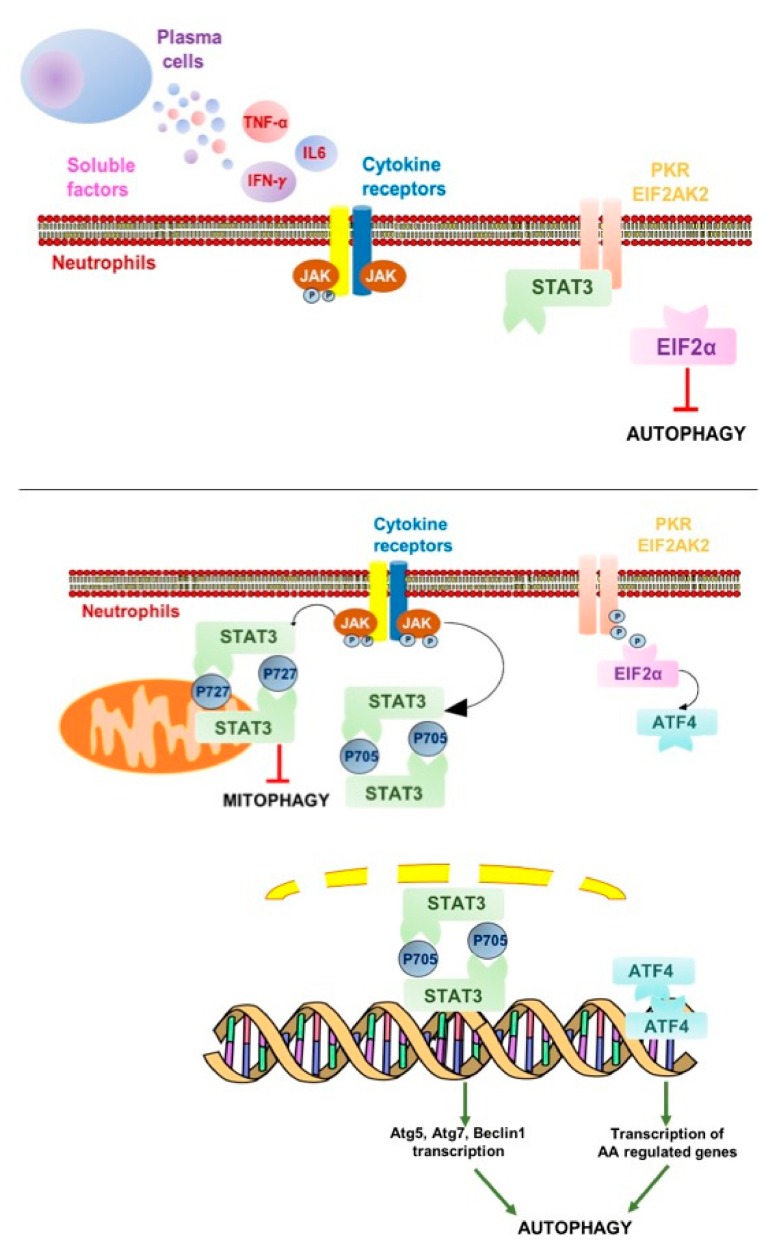
STAT3: Signal transducer and activator of transcription 3; ATG: Autophagy related; EIF2α: Eukaryotic translation initiation factor 2; ATF4: Activating transcription factor 4; PK3: Protein kinase RNA-activated also known as protein kinase R. Green arrow: activation; Red T-bar: inhibition. Proposed role of the JAK-2/STAT3 activation pathway in myeloma high-density neutrophils. Soluble factors released by MM plasma cells (i.e., interleukin1β, tumor necrosis factor α, and interleukin 6) or other infiltrating immune cells (i.e., interferon γ) can trigger the JAK/STAT pathway in surrounding myeloid cells. Specific residues in the cytoplasmic domain of the transmembrane receptors are recruited and phosphorylate STAT3. Therefore, phosphorylated STAT3 (pSTAT3-705) homodimerizes and can translocate to the nucleus where it regulates autophagy-related gene expression. In other cell types, to preserve their function and integrity, phosphorylated STAT3 (pSTAT3-727) homodimers translocate to the mitochondria to inhibit mitophagy, but this branch of the pathway is not relevant in HDNs that have only a few mitochondria, making them highly sensitive to autophagy triggering in the presence of specific soluble factors released by neoplastic cells.

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
