# Peer review of "Plasticity of High-Density Neutrophils in Multiple Myeloma is Associated with Increased Autophagy Via STAT3"

_ijms, 2019, doi:10.3390/ijms20143548_

Round 1
Reviewer 1 Report
In this manuscript, Puglisi et al showed the autophagy in neutrophils isolated from multiple myeloma (MM). Those neutrophils showed progressive autophagy with increased ROS generation and serum from MM induced autophagy in healthy neutrophils. They further identified that autophagy in neutrophils induce STAT3-pS727 expression resulting in increased IFN-γ expression. While this manuscript is of general interest for Int. J. Mol. Sci., there are major and minor points that need to be addressed.
Major concerns
1. Methodologies for measuring autophagy in neutrophils:
There are concerns regarding methodologies of measuring autophagy in neutrophils. The authors assessed the autophagic status of neutrophils by simply quantifying autophagic vacuoles (Figure 2 and 3). Although they determined the fluorescence levels of LC3, I concerned that this quantification is not enough for persuading readers. Current consensus recommends multiple assays for monitoring autophagy [PMID: 26799652]. One of the important indicators is LC3. The author discovered increased LC3 fluorescence in neutrophils isolated from MM (Figure 2D). However, guideline recommend quantifying not only overall increase of LC3 fluorescence but also conversion of LC3-I into LC3-II. Moreover, degradation of p62 is used to assess normal autophagic flux. I believe that there are many difficulties for handling patient samples. However, I believe that the sole quantification of autophagic vacuoles are not sufficient. Therefore, the quantification of LC3 with p62 at protein level should be addressed.
2. The role of autophagy in neutrophils:
As the authors mentioned (p.2, line 81-83), Autophgy is important for shifting metabolism from glycolysis toward oxidative phosphorylation during neutrophil differentiation [PMID: 28916263]. However, recent advances in neutrophil research revealed that autophagy is important for effector functions of neutrophils such as neutrophil extracellular trap (NETs) formation [PMID: 21060338, 23720022, 28358992]. Although authors identified ROS generation in MM neutrophils, the quantification of NETs formation in those neutrophils should be addressed.
Minor concerns
1. The methods for measuring grains in neutrophils (Fig. 1C) was missing.
Although Fig. 1C shows increased granule components in MM neutrophils, methods are not provided in the manuscript.
2. Legends for arrows in Fig. 2A was missing.
3. Specify each color in Fig 2D.
4. Fig 3B, the authors claimed bafilomycin reversed the effects of MM serum on healthy neutrophils. However, the legend indicates that the effects of bafilomycin on healthy, untreated neutrophils was examined. Also, it should be clarified why the authors treated healthy neutrophils with MGUS serum and MM serum simultaneously. I am not sure whether this is just misspelled.
Author Response
Dear Reviewer 1,
thanks a lot for your suggestions. Despite the little time-frame given to complete the revision (10 days) and the availability of samples obtained from 3 newly-diagnosed patients, we addressed your concerns as follows:
1. We quantified LC3B conversion and p62 amount in neutrophils freshly sorted from MM patients, treated for 4 hours in presence of bafilomycine, using both western blots and immunofluorescence, as reported in Figures 3-4.
2. We clarified in the Methods Section the quantification of ROS generation in MM neutrophils by flow cytometry, and we added the quantification of NETs formation in Figure 2L.
3. We added a paragraph in the methods section to to describe how we measured autophagosomes by electron microscopy, and we clarified the point in the legend for Figures 1-2
4. We removed the past version of Fig. 2A
All Figures and results description have been changed according to your suggestions to improve the quality of the manuscript.
Thanks again,
Dr. Romano
Reviewer 2 Report
The article by Puglisi et al. examines how factors in serum from Myeloma patients are activating neutrophil autophagy and Jak/Stat signaling. They first describe an altered phenotype of neutrophils isolated from the blood of those patients, characterized by hypolobulated nuclei, altered granule content and increased survival. In this presumably heterogeneous HDN population, signs of autophagy are increased, and this can be replicated in healthy HDN by myeloma patient serum supplementation. Evidence for activated Jak/stat and autophagy in patient neutrophils and in healthy neutrophils in response to patient serum is presented, and acidic vesicular organelle numbers are restored after treatment with a Pan-Jak inhibitor.
The article is timely and relevant, well written, and addressing clinically important aspects of immune impairment and -response in the context of multiple myeloma. Although in some experiments transformed cell line supernatants are used, most of the work is performed in primary patient cells and sera.
The data would benefit from additional controls and clarifications, specifically:
Figure legends lack experimental details such as number of experiments, replicates, patients.
Figure 1:
- The HDN fraction does not seem to express CD14 in the representative gating.
- Is this from healthy donor? it is crucial to determine % purity and marker changes in the samples from patients too. This is essential to address if the phenotypic neutrophil changes represent appearance of MDSC or other altered subsets of neutrophils, or if this represents a change in the existing, homogenous population of neutrophils that is shown in Fig 1A. The % and absolute numbers of these cells in Control and Patients would be relevant, also to understand potential population effects in all later assays that use the entire HDN fraction.
- When the CD14+CD15+ neutrophils would be sorted from patients and controls, do they contain the immature-like, longer-lived, more granular cells, and do these cells show altered surface markers?
- The counting of “grains” from the presented relatively low resolution images appears very difficult. Does this represent antimicrobial granules? They could be quantified better from the EM data that is available. An Mpo-IHC staining should help better quantify granule content, or qPCRs for granule genes.
- Blind counting or software-driven automatic counting should be used for granule content, nuclear lobularity and vesicle counts.
- 60% of neutrophils surviving after 7 days of culture in absence of cytokines is a stunning phenotype of the MM HDN, this would be worth resolving in a timecourse and with a more sophisticated setup. Details on the survival assay are missing, number of donors etc. A good way to do this is quantification of live cells e.g. by LDH assay as a timecourse from separate cultures, ideally separate donors/patients)
- The probe that is used to determine ROS is not mentioned in Fig 1E.
Figure 2:
- How are acidic vesicles determined by EM? Morphologically, lysosomes, autophagosomes, autolysosomes, endosomes, golgi vesicles, etc all appear extremely similar by TEM. Is this based on the presence of double membranes on autophagosomes? More detail required in figure legends. Small circular “vesicles” close to the cell membrane can represent cutting artefacts. An immunolabeling for LC3 would be gold standard, if this is too difficult, a two hour incubation with bafilomycin will block autophagosome degradation, sothat the buildup of autophagosomes can be quantified and represents “autophagic flux”.
- Acridine orange alone is not an ideal marker to measure autophagosomes, there are good assays for flow cytometry available: Klionsky et al., Guidelines for monitoring autophagy.
- LC3 immunostaining, as in Fig2D, is a good assay, but it does not represent autophagic flux, as claimed in the text. For this, a drug like Bafilomycin must be used to measure the buildup of autophagosomes over time. A good staining should allow to quantify individual autophagosome foci, rather than fluorescence MFI.
Figure 3:
- The same limitations in the measurement of autophagy apply as in Figure 2.
- Which markers are used to gate on neutrophils before showing acridine orange MFI? This would be extremely interesting to see if there is a subset of cells in patients that can be distinguished by surface markers and has particular high autophagy levels? From Fig. 1A it appears there is a subset of neutrophils with normal morphology, mixed with a hypolobulated population in the patients. The large error in lobularity counts similarly suggests some hypolobulated cells among normal neutrophils in myeloma context. Is the hypolobulated subset the one that has higher autophagy?
- In Fig 3B, which serum received Bafilomycin? Bafilomycin inhibits acidification, sothat acridine orange staining is lost, but autophagosomes should build up in response to the drug.
Figure 4:
- In contrast to Figure 3, this data is pre-gated on CD15+ cells, why the difference?
In summary, while the study is well conceived, relevant and conceptually meaningful, there is a lot of potential to improve the presentation and description of the data, and data itself, with the essential points to be addressed:
- Surface markers and percentages of healthy and patient HDN fraction to explore the potential of identifying the morphologically changed subset, as described in figure1, by surface markers.
- Improving the techniques to detect and measure autophagic flux. LC3 western blots +/- flux inhibitor (Baf) remain the best practice, or LC3 immuno-EM +/- Baf, LC3 flow cytometry kits or similar, as described in Klionsky et al.
Author Response
Dear Reviewer 2,
thanks a lot for your suggestions. Despite the little time-frame given to complete the revision (10 days) and the availability of samples obtained from 3 newly-diagnosed patients, we improved the quality of figure legends adding experimental details such as number of experiments, replicates, patients and we addressed your concerns as follows:
Figure 1:
You are right, indeed the HDN fraction does express CD14; there was a mispelling now corrected.
In the previous figure 1 the gating strategy shown was derived from a MM patient. As specified in the text and percentage purity was assessed in each sample. However the strategy of isolation has been clarified in the new version of figure 1.
MDSC can be defined in the upper-ring after Ficoll-based separation. In whole peripheral blood, granulocytic-like MDSC can be identified as CD15+Lox1+ cells (Condamine 2016). In the new Figure 2 Figure we clearly show that there is a fraction of CD15+Lox1+ cells in both MM- and healthy-derived HDNs,<5%.< p="">
On the other hand, morphology shows clearly that HDNs are not homogenous, but based on MDSC definition, cells we are isolated cannot be defined MDSCs. After the revision, in figure 2 reported the percentage of CD15+Lox1+ PMN-MDSC in peripheral blood of healthy and MM patients. These changes should help to clarify potential population effects in all later assays that use the entire HDN fraction.
When the CD14+CD15+ neutrophils would be sorted from patients and controls, do they contain the immature-like, longer-lived, more granular cells, and do these cells show altered surface markers?
Author response: The phenotype of high-density neutrophils is the same for patients and controls, identified as CD66b+CD15+CD14- cells in the fraction of the bottom cells. Please look at the new version of Figure 2 and the comments reported above.
The counting of “grains” from the presented relatively low resolution images appears very difficult. Does this represent antimicrobial granules? They could be quantified better from the EM data that is available. An Mpo-IHC staining should help better quantify granule content, or qPCRs for granule genes.
Since the confusion generated from images arrangement in panels, the new version of Figures 1-2 address more clearly the morphology changes occurring in controls and patients, and legends have been improved accordingly.
We performed a timecourse to measure the MM-HDN survival, not shown in the previous version of the manuscript. In the new Figure 2 we included images from AnnexinV/PI essay, separated for representative donors and patients cultures, at 3 different timepoints (24-48-96 hours), with the relative LDH assay and quantification.
The probe used to determine ROS has been mentioned in Fig 2, and more details about ROS quantification by flow cytometry have been added in the methods section.
Unfortunately, we could not test alternative assays for flow cytometry to measure autophagosomes, but we used other techniques to confirm the autophagy changes in primary samples, based on WB and immunofluorescence, as shown by new Figures 3-4.
Acidic vesicles have been determined based on the presence of double membranes on autophagosomes, as detailed in the legends of new Figures 1-3. In the new Figure 3 we showed autophagic flux as detected by immunofluorescence in both healthy and MM-HDN cultured with BAF for 3 hours.
We repeated LC3 immunostaining shown in the former Fig2D, culturing healthy and MM-HDN in presence of Bafilomycin to measure the buildup of autophagosomes over time and to quantify individual autophagosome foci, rather than fluorescence MFI.
Similarly, in the former Figure 3, now modified in Figure 4, we report both the changes in the number of autophagosome foci and avo-test percentage.
are used to gate on neutrophils before showing acridine orange MFI were CD11b, CD15 and CD14, as clarified in the legend of Figure1. For technical reasons, we could not see if there is a subset of cells in patients that can be distinguished by surface markers and has particular high autophagy levels.
Figure 4:
Question: In contrast to Figure 3, this data is pre-gated on CD15+ cells, why the difference?
Author response: All data were pre-gated on CD15+ cells, but in the previous version of the manuscript this detail was not clearly shown.
Question- Surface markers and percentages of healthy and patient HDN fraction to explore the potential of identifying the morphologically changed subset, as described in figure1, by surface markers.
Author response: we included in the new version of Figure 2 all details about the detection of MDSC and HDN in peripheral blood of controls and patients.
Question: Improving the techniques to detect and measure autophagic flux. LC3 western blots +/- flux inhibitor (Baf) remain the best practice, or LC3 immuno-EM +/- Baf, LC3 flow cytometry kits or similar, as described in Klionsky et al.
Author response: according to your suggestions we measured autophagic flux by LC3 western blots and LC3 immuno-EM +/- Baf as reported in the new Figures.
Round 2
Reviewer 1 Report
The authors answered all my concerns.
Author Response
Dear Reviewer 1,
thanks a lot for your suggestions.
Reviewer 2 Report
The manuscript by Puglisi et al. has benefitted substantially from the extensive revisions and re-organization particularly of Figures 1-3, resulting in a significantly improved manuscript. Many of the missing details have been filled in, particularly in terms of methodology/data acquisition. I suggest a few minor changes before publication.
- Fig. 1 FACS graphs showing the CD15 expression in the HDN fraction should not cut off the CD15 negative events. Negative event should be less than 10% as described in text.
- Fig. 2L This new data claims increased NET production based on DAPI-staining via microscopy. Without quantification and NET specific assays such as citrullinated Histone or similar, this conclusion cannot be drawn. I would suggest to improve or remove this non-essential data.
- P3 line 102 CD15 LDN should say CD15 HDN
- Bafilomycin is misspelled throughout.
- P3 L 118 should state that both conditions are higher THAN healthy
- P3 L 129 The stated 169% do not match up with the figure shown, should be about 140%.
Methods section 4.9 ROS detection lacks any text.
Author Response
Dear Reviewer 2,
thanks for your new suggestions.
In the revised version of the enclosed manuscript we added a panel in Figure 1 to show the CD15 negative events.
In Figure 2, we left the panel L about NET production, as requested by Reviwer 1, but we removed the statement about any comparison between healthy and MM HDNs. This non-essential data is useful to highlight that at 48 hours both healthy and MM HDNs are alive and functional, while changes in survival appear later, enforcing data shown in the time-course.
Mispellings and syntax have been reviewed and corrected, and description for ROS detection in the methods section 4.9 has been added.
Please find highlighted in green the changes of this version.
Best regards,
Dr. Romano